# Post-Weld Heat Treatment Effects on Mechanical Properties and Microstructure of AA6061-T6 Butt Joints Made by Friction Stir Welding at Right Angle (RAFSW)

**Mahboubeh Momeni and Michel Guillot ***

PI2/REGAL Research Team, Department of Mechanical Engineering, Laval University, Quebec, QC G1V 0A6, Canada; mahboubeh.momeni.1@ulaval.ca

\* Correspondence: mguillot@gmc.ulaval.ca; Tel.: +1-(418)-998-6549 or +1-(418)-656-3343

**Abstract:** Friction stir welding (FSW) provides users with many advantages over fusion welding techniques. Nevertheless, it is not widely employed in current industry mainly due to high equipment costs and royalties. To overcome these issues, a low-cost FSW technique operated at a right angle, called RAFSW, has recently been developed by our research team. To make the RAFSW technique reliable for potential users, we are going to analyze the effect of various post-weld heat treatments (PWHT) on the mechanical and physical properties of the RAFSW joints. To this end, optimized process parameters are used to weld butt joints of an AA6061-T6 alloy. The joints were characterized using a tensile test, a micro-hardness test, and metallography techniques. The most efficient aging time was obtained for various aging temperatures. Moreover, it was found that artificial aging at 220 °C for 30 min could be used as a fast and cost-effective artificial aging PWHT for the industrial sector. In addition, the repeatability of the PWHTs were demonstrated by studying the effect of waiting time prior to the artificial aging. Finally, it was revealed that a single fast artificial aging process is more beneficial than solubilizing followed by an artificial aging process in terms of tensile properties, consumed time, and cost.

**Keywords:** friction stir welding at right angle; post-weld heat treatment; aging; solubilizing; tensile properties; hardness

## 1. Introduction

Friction stir welding (FSW) has attracted an increasing interest among researchers and industrial sectors over the past two decades [1]. Compared to the common fusion-welding techniques, like MIG welding, FSW provides users with many advantages such as better mechanical and fatigue properties of the weld, the ability to weld dissimilar alloys and non-weldable alloys by fusion welding techniques, less energy-consumption, and environmental friendliness [1–7]. However, there are some obstacles to its prevalent use such as the need for CNC equipment, the large forces involved, the need to use sturdy fixtures, and, generally, the equipment costs and royalties [2,3,8]. To have a cost-effective FSW process, the PI2/REGAL team at Laval University has recently developed a low-cost FSW technique operated at a right angle [9–14]. In this technique, called RAFSW, common low-cost 3-axis CNC machines, which are used for machining, are utilized for the RAFSW technique too, without any modification of the CNC machines.

Another barrier against the widespread use of FSW in industry is the fact that there are a lot of uncertainties, scientific gaps, and lack of guidelines regarding other aspects of this process, like the post-weld heat treatment (PWHT) of FSW joints [2]. Though it is confirmed that PWHTs could greatly



enhance the mechanical properties of the FSW joints [15–17], the reported research is limited. Indeed, there is lack of research on the effect of PWHTs on the FSW joints, especially for welded samples by FSW at a right angle and at high traverse speeds. Therefore, it is essential to shed light on the impact of various PWHT processes on the optimized welded samples by the RAFSW technique to make it reliable for potential industrial users. Various parameters affect the physical and mechanical properties of FSW joints such as microstructure, tensile properties, and the micro-hardness distribution of the weld area [6,18,19]. Tool design is one of the influential parameters since the mixing and stirring mechanisms within the weld area are in control of that [20]. Moreover, the generated forces during the process and the process parameters including the tool tilt angle, plunge depth, rotational speed, and traverse speed affect the physical and mechanical properties of the joint, because they control mixing and stirring processes, the amount of generated heat, and the cooling rate during the FSW process [1,6]. The developed RAFSW technique has its own features in terms of tool design, involved forces, and optimized process parameters. Therefore, the physical and mechanical properties of the welds are affected by all of them [14]. Thus, it is necessary to investigate and characterize the effect of different natural aging, artificial aging, and solubilizing processes on the RAFSW joints to make the process reliable for industrial use.

Therefore, the main objective of the present paper is to study the effect of various PWHTs on the mechanical and physical properties of the RAFSW joints welded under optimized process parameters at high traverse speeds obtained in our previous research [14]. To make the welds, we have utilized a recently developed RAFSW technique using low-cost 3-axis CNC machine tools [9–14].

## 2. Materials and Methods

### 2.1. Preparation of the Joints

In this research, the RAFSW technique was employed to butt-weld extruded AA6061-T6 flat bars that were 6.35 mm thick, 76.2 mm wide, 250 mm long, and cut by a power hacksaw. Based on the extruder specification for the as-received AA6061-T6 alloy, it had a nominal chemical composition of 0.83 wt.% Mg, 0.55 wt.% Si, 0.19 wt.% Cu, 0.19 wt.% Fe, 0.05 wt.% Cr, and 0.07 wt.% Mn; the rest was aluminum. Tensile test results of the as-received material showed that the ultimate tensile strength and ductility were 285 MPa and 16.4%, respectively. To make the joints, two flat bars were fixed onto a rigid back-plate. The back-plate was placed on a 3-axis Kistler 9265B dynamometer fastened to the table of a 3-axis CNC machining center. The utilized 3-axis CNC machine was a Fryer MC-15, which had a 25 HP spindle using CAT40 tool holders with a maximum rotational speed of 8000 rpm and an axis peak trust of 15 kN. To conduct RAFSW, a tool designed specifically for this purpose was mounted into a long reach tool holder. The shoulder diameter of the tool and its pin length were 15 and 6 mm, respectively. The pin shape is spiral and some grooves were designed on the shoulder. Figure 1a,b demonstrate the mentioned installation, the utilized tool, and the tool holder. The described set-up and installation is exactly what was used in the previous research regarding the optimization of the RAFSW technique [14]. The obtained optimized process parameters were used to weld the samples according to our previous research [14]. The tool plunge depth, rotation speed, and traverse speed were 6.15 mm, 3500 rpm, and 540 mm/min, respectively [14]. Figure 1c depicts the top view of the single-pass welded joint along the extrusion direction of the bars under the mentioned process parameters.

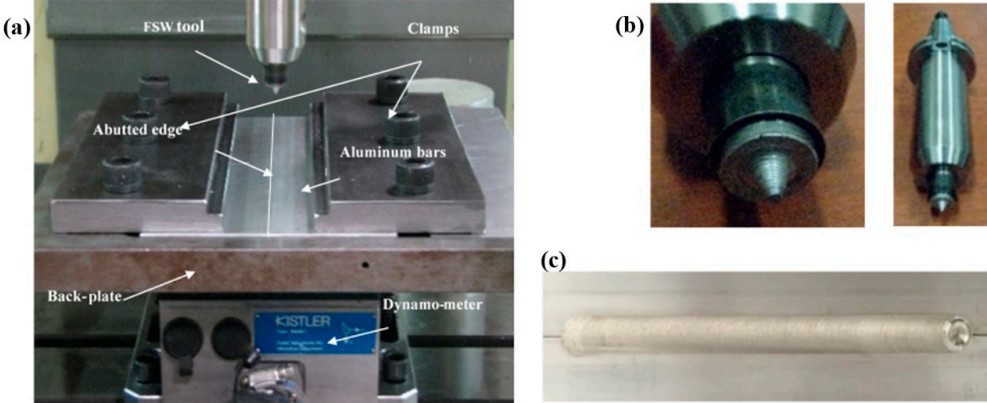

**Figure 1.** (**a**) The set-up of welding consist of the extruded bars, the clamping system, and the Kistler 3-axis dynamometer; (**b**) friction stir welding (FSW) tool and tool holder; (**c**) top view of the welded sample by optimized process parameters [12].

### 2.2. Post-Weld Heat Treatment

A calibrated PYRADIA furnace equipped with a monitoring system for temperature variations was used for PWHT processes. The targeted PWHTs were different combinations of the natural aging, artificial aging, and solubilizing processes. The effect of natural aging time on as-weld samples was studied, firstly. Then, the impacts of various artificial aging processes were characterized at 160, 180, 200, 220, and 240 °C for various periods from 20 min to 18 h on the as-weld samples. Moreover, the effect of the natural aging prior to the artificial aging was studied to evaluate the repeatability of the PWHT results when there is a delay between welding and the artificial aging process. Finally, the effect of the solubilizing was studied at 530 °C for 1 h, followed by the optimized artificial aging process called T6 obtained from the previous set of experiments. For each PWHT process, two samples—one welded sample and one raw material sample—were subjected to the designed heat treatments.

### 2.3. Characterization

A tensile test was done to evaluate the tensile properties of the joints. The uniaxial tensile test was done according to the standard of American Society for Testing of Materials, ASTM E8M-04 standard. The geometry of the machined samples is illustrated in Figure 2. For that, a hydraulic testing machine with a 44.5 kN load cell was used. The crosshead speed of the testing machine was 1 mm/min. Moreover, a high-resolution Epsilon extensometer was used during the tests. For metallographic investigations, welds were cut around the joint area and mounted in the phenolic resin. Then, the mounted samples were polished to 2000 grits SiC papers followed by a final polishing by colloidal silica suspension using an automatic Buehler polishing machine. Afterwards, the polished samples were pre-etched with a 10 wt.% NaOH solution followed by etching with Weck's reagent [14,21]. Then a Nikon K100 optical microscope was used to take the micrographs. The macrograph of the joint was captured using a high-resolution camera. The used macro-etchant was a solution containing 75 mL of $HCl^+$, 25 mL of $HNO_3^+$, and 5 mL of HF diluted with 25% distilled water. Consequently, a Buehler Vickers micro hardness-testing machine was used to measure the micro hardness of the samples within and around the joint. The measurements were done according to the ASTM E384 standard by applying a 300 g load and a dwell time of 15 s.

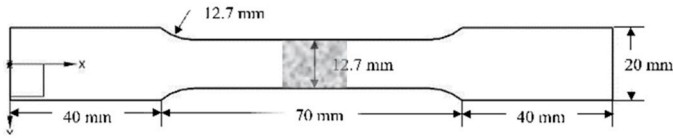

**Figure 2.** Schematic of tensile test samples.

## 3. Results and Discussion

### 3.1. Natural Aging

Figure 3 depicts the effect of natural aging time on the tensile properties of the as-welded RAFSW samples. It indicates that the tensile properties drastically changed until seven days of natural aging. Moreover, it can be inferred that the tensile properties were almost stable after 14 days of natural aging. The tensile strength increased by passage of time while the ductility decreased. This is attributed to the kinetic of the precipitation in different regions of the joint. The precipitate evolution led to partial recovery of strengthening through the welded area [22]. As a result, the tensile strength of the RAFSW joint boosted in the course of time. Generally, the stronger the joint, the less ductility it will have, which is confirmed with our results as shown in Figure 3. The stabilized tensile strength, ductility, and joint efficiency of the as-weld sample were 204 MPa, 14.2%, and 72%, respectively.

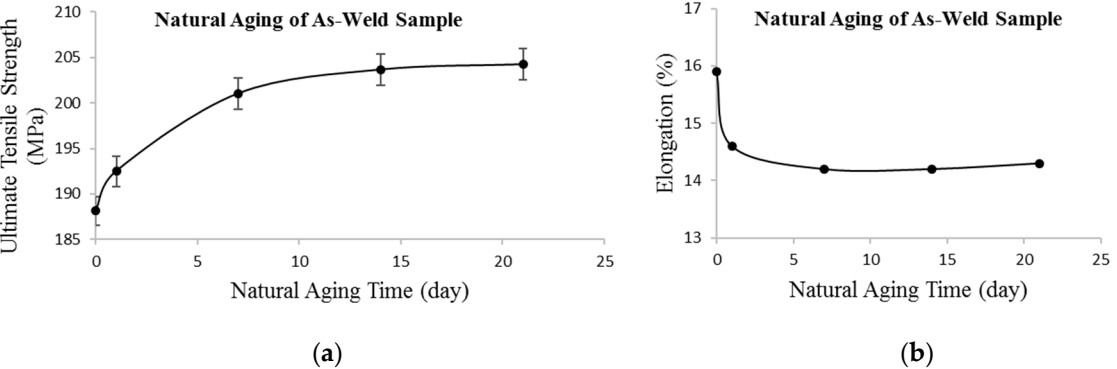

(**a**)                                                                                    (**b**)

**Figure 3.** The effect of natural aging time on tensile strength: (**a**) And elongation; (**b**) of welded samples at optimized process parameters by right angle friction stir welding (RAFSW).

Figure 4 demonstrates the macrostructure and micro-hardness map of the naturally aged RAFSW sample. Distinct regions of FSW welds called the nugget zone (NZ), the thermo-mechanically affected zone (TMAZ), the heat-affected zone (HAZ), and the unaffected base material (BM) are identified in Figure 4a. The micro-hardness map across the weld area, shown in Figure 4b, is in conjunction with the identified zones in the macrostructure shown in Figure 4a. Generally, the hardness drops in the weld area for the FSW joints of AA6061-T6, as can be seen in Figure 4b. This behavior is mainly associated with the dissolution or coarsening of the precipitates within the weld area during the welding process and the re-precipitation and coarsening of the precipitates during the subsequent aging process [22]. Figure 5 demonstrates the microstructure around the NZ. The borders of the NZ, TMAZ, and HAZ are obvious in the advancing side, while they are not distinguishable in the retreating side of the RAFSW samples. This asymmetric behavior originates from the asymmetric effect of the rotating tool inside the material [23].

As can be seen in Figure 6a–c, the NZ shows a fine, equi-axed microstructure compare to the BM because of dynamic re-crystallization and the formation of special grain boundaries during the FSW process [24]. The microstructure of the NZ changes from top to the bottom of the nugget zone. The finest grains are observed in the top and the biggest in the middle near the bottom, as seen in Figure 5a–c. This could be attributed to both mixing and stirring mechanisms in different areas of the NZ and the difference in the amount of heat input which is more in top-side due to the contact with the tool shoulder. Moreover, the difference in cooling rate plays a role in the final microstructure of the grains and the precipitates evolution during and after the welding process [25]. The finer the microstructure, the higher the micro-hardness. In this connection, the microstructure observations for the NZ are in conjunction with the micro-hardness profile shown in Figure 4b. From Figure 6d–f, it is obvious that the grain structure in the HAZ area far from the NZ seems like BM, while in the HAZ near the NZ, the grains are relatively smaller and less elongated than the HAZ far from the NZ. This could be originated from the difference in experienced thermal cycle during the weld. The micro-hardness

map (Figure 4b) demonstrates that the HAZ regions far from the NZ have much higher hardness than the HAZ regions near the NZ area. This could be assigned to the major difference in the morphology, size, and distribution of the precipitates in these areas.

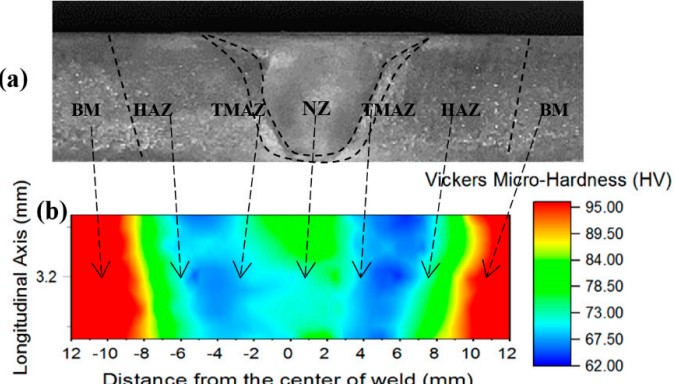

**Figure 4.** (**a**) Macrostructure at the cross-section of the joint marked with the distinct weld regions; (**b**) Micro-Vickers hardness map around the welded region of a naturally aged sample made by the RAFSW technique operated at optimized process parameters.

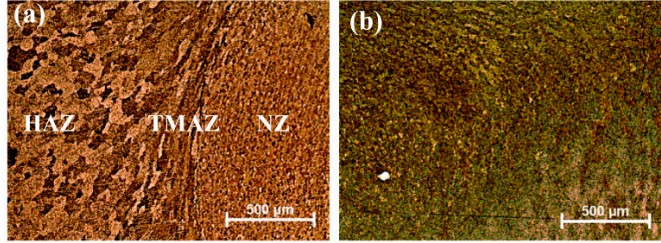

**Figure 5.** Microstructure of the welded sample around the nugget zone at: (**a**) Advancing side; (**b**) and retreating side for naturally aged joints made by RAFSW at optimized process parameters.

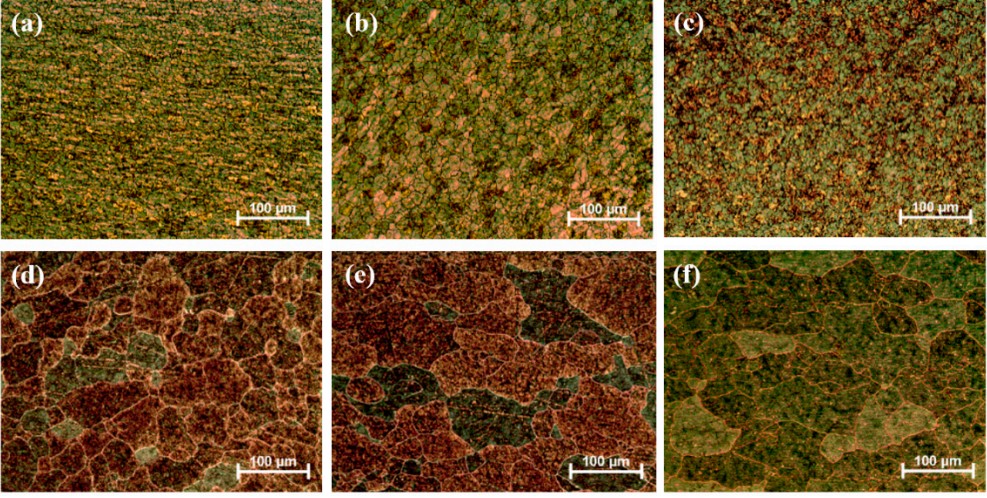

**Figure 6.** Microstructure of: (**a**) Top; (**b**) middle; (**c**) and bottom of the nugget zone (NZ); (**d**) base material; (**e**) the heat-affected zone (HAZ) area far from the weld region; (**f**) and the HAZ area near the NZ for naturally aged joints made by RAFSW.

### 3.2. Artificial Aging

There is lack of study on the effect of artificial aging at different times and temperatures for FSW samples, specially at the zero-tilt angle and high traverse speeds. Meanwhile, the industry is in need of reliable, cost-effective PWHTs. Therefore, in this part of research, different artificial aging processes

on RAFSW butt-joints are examined. In this regard, Figure 7 demonstrates the effect of artificial aging time and temperature on the tensile strength of the RAFSW-welded samples. Generally, the higher aging temperature, the lower time is needed for precipitates evolution [26]. It is obvious that, at first, the tensile strength of the samples increases with aging time at each aging temperature. This originates from the strengthening mechanism of evolved precipitates in the weld area due to the aging process. As can be seen in Figure 7, after reaching a peak of tensile strength at a certain aging time for a given aging temperature, the tensile strength starts to decrease due to over-aging [26]. As is observable, this reduction is sharper for higher aging temperatures due to the fast kinetics at higher temperatures. It can be concluded that the highest joint efficiency and tensile strength under aging at 180 °C for 18 h are 90% and 257 MPa, respectively. The results of the artificial aging at 160 °C for 18 h were approximately similar. The tensile strength in this case was as high as 254 MPa. Since the industry is in need of fast and cost-effective PWHTs, one can conclude that a fast, cost-effective artificial aging process at 220 °C for 30 min is quite beneficial. In this case, the joint efficiency and tensile strength of the RAFSW joints would be as high as 85% and 241 MPa, respectively. Table 1 provides a summary on the tensile properties of the plain material, naturally aged weld, and artificially aged welds at optimized conditions obtained from Figure 7. In macro-scale, the elongation of the samples is shown in Table 1.

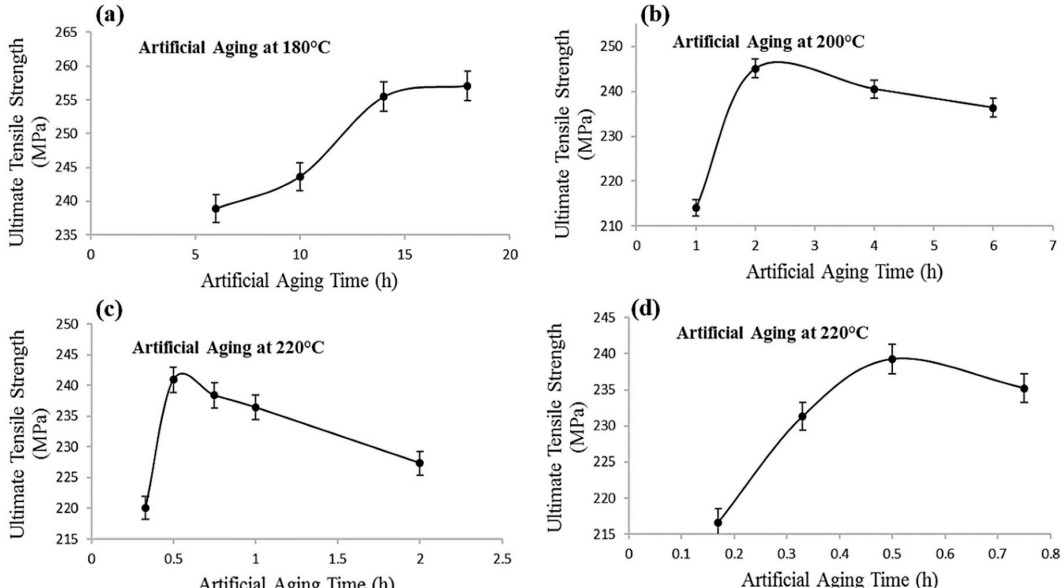

**Figure 7.** The effect of artificial aging at: (**a**) 180 °C; (**b**) 200 °C; (**c**) 220 °C; (**d**) and 240 °C under different aging times on the tensile strength of the RAFSW samples.

**Table 1.** Tensile properties and joint efficiency of the welds aged naturally and at optimized artificial aging conditions.

| Samples Condition | UTS (MPa) | El (%) | Joint Efficiency [2] |
|---|---|---|---|
| Plain Material (as received) | 285 | 16.4 | |
| As-weld | 188 | 15.9 | 66% |
| NA [1] for 14 days | 204 | 14.2 | 72% |
| AA [1] at 160 °C for 18 h | 254 | 4.1 | 89% |
| AA at 180 °C for 18 h | 257 | 4.2 | 90% |
| AA at 200 °C for 2 h | 245 | 6 | 86% |
| AA at 220 °C for 30 min | 241 | 6.1 | 85% |
| AA at 240 °C for 30 min | 230 | 6.6 | 81% |

[1] NA and AA are abbreviations for naturally aged and artificially aged, respectively. [2] Joint efficiency is the ratio of the strength of the joints to the strength of the plain material in percentage.

Figure 8 demonstrates the micro-hardness distribution at centerline through the weld area of a naturally aged joint, artificially aged at 160 °C for 18 h, and recommended fast aged sample at 220 °C for 0.5 h. Indeed, during the subsequent artificial aging process on welded joints, the dislocations density and the amount of solute for precipitation differed in different zones of the weld. These factors can mainly control the re-precipitation kinetics. Moreover, the time and temperature of the aging process affect the precipitation evolution in different zones [27]. As a result, the amount of hardness recovery in different zones of the weld would be different. Based on these facts, it can be seen that the naturally aged sample shows a normal w-shape distribution of hardness, as shown in Figure 8. The minimum hardness is within the HAZ area near the TMAZ. This is in accordance with the minimum recovery of hardness by precipitates hardening in this area [22]. The NZ area demonstrates a higher hardness, which could be due to its finer microstructure than the HAZ area located near the NZ region. However, in both areas, the precipitation hardening mechanism is mostly gone due to the dissolution or coarsening of precipitates under thermal cycles of the FSW process [22]. From Figure 8, it can also be concluded that an artificial aging process at 160 °C for 18 h leads to the recovery of hardness. It could be assigned to precipitate-strengthening mechanisms by re-precipitation and precipitate evolutions [17,28]. In the NZ region, the hardness is considerably restored, owing to the effective type, size, morphology, and homogenous distribution of the precipitates evolved during the re-precipitation process by aging [29]. This restoration of hardness partially occurred in the TMAZ and less in the HAZ area near the NZ, as shown in Figure 8. It is obvious that artificial aging at 220 °C for 30 min yields less hardness recovery than artificial aging at 160 °C for 18 h. This could be due to the differences in the number, size, morphology, and distribution of the evolved precipitates, because the evolution of the precipitates is a diffusion based mechanism, which is significantly affected by the time and temperature of the aging process [17,26].

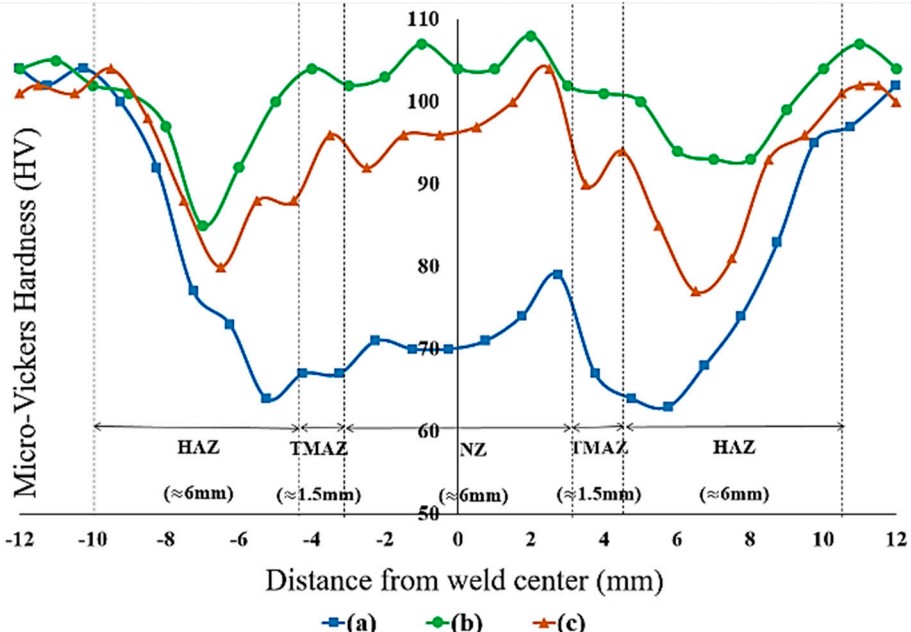

**Figure 8.** Hardness distribution along the centerline of the cross-section of the joints for: (**a**) Naturally aged RAFSW sample; and for artificially aged samples at: (**b**) 160 °C for 18 h; and (**c**) 220 °C for 30 min.

Figure 9 illustrates the microstructure around the NZ area of the RAFSW joint aged artificially at 160 °C for 18 h. It is evident that the transition between distinct weld zones is more distinguishable in this sample compare to the naturally aged sample, as seen in Figure 5. Microstructural observations indicate there is no significant difference in the grain structure of the best artificial aged samples at each aging temperature, including 160 °C for 18 h, 180 °C for 18 h, 200 °C for 2 h, 220 °C for 30 min, and 240 °C for 30 min. They all show a similar pattern under optical microscope, like what we have seen in

Figure 10. Indeed, the difference in their mechanical properties is originated from the difference in the type, size, morphology, and distribution of the precipitates evolved in their different weld zones during the artificial aging process. In addition, the microstructure of the naturally aged sample shown in Figure 6 looks like the artificial aged microstructure of Figure 10, which means no considerable change happens in the shape and size of grains by the conducted artificial aging processes.

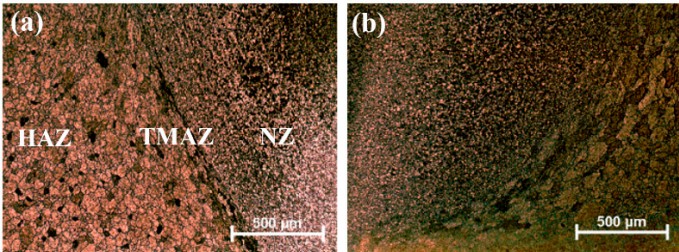

**Figure 9.** Microstructure of welded sample around nugget zone at advancing side (**a**) and retreating side (**b**) for the RAFSW joint aged artificially at 160 °C for 16 h.

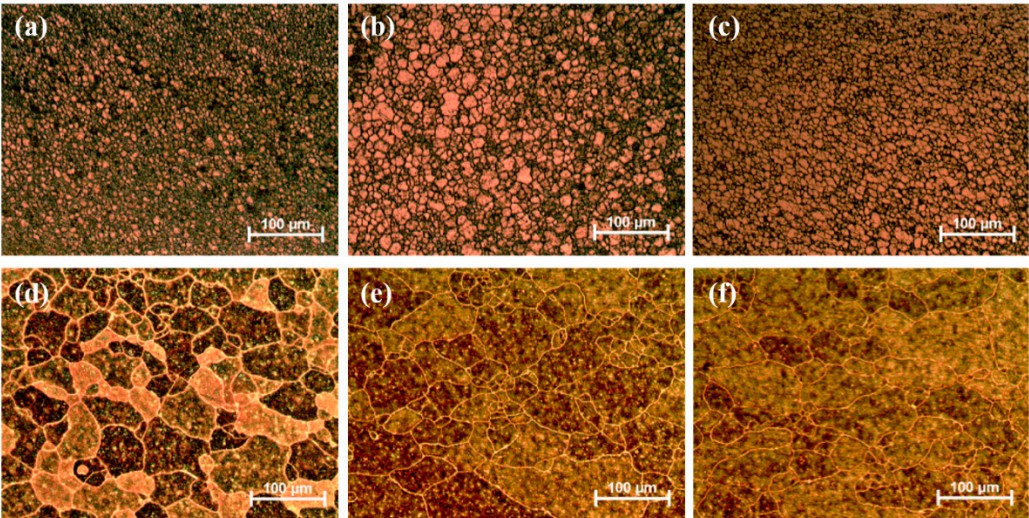

**Figure 10.** Microstructure of top (**a**), middle (**b**), and bottom (**c**) of the NZ zone, base material (**d**), the HAZ area far from the weld region (**e**), and the HAZ area near the NZ (**f**) for the RAFSW joint aged artificially at 220 °C for 30 min.

Based on Table 1 and Figure 8, although artificial aging processes boost the strength of the joints and the hardness in the weld zone, it cannot recover the elongation of the weld samples. This can be attributed to the differences in the microstructure of the artificially aged joint compared to plain material. Moreover, it can be assigned to the inconsistency in the microstructure of different zones of the artificial aged weld samples, as shown in Figure 10. Moreover, the size, morphology, and distribution of the precipitates are affected by the experienced thermal cycles during and after the welding process. Precipitates act as barriers against movement of dislocations during plastic deformation [22–25]. Thus, it is expected that different zones of the welds show different local mechanical properties such as elongation and plasticity [22,30]. It has been reported that indentation micro-hardness results demonstrate a good correlation with local mechanical properties of the weld zones [18,30]. According to Figure 8, the indentation results from the micro-hardness diamond probe demonstrate the difference in the behavior of weld zones. In all samples, the diamond probe had left the biggest indentation prints in the HAZ area, which can indicate that the material in the HAZ area demonstrates more plasticity than other zones. This characteristic of the HAZ makes it prone to develop necking and breakage during the tensile test [18,19]. This conclusion is in agreement with our experimental results, as all aged joints necked and broke in the HAZ area. Moreover, the identified relationship between the

hardness and toughness of the FSW samples can be noted. Indeed, the less hardness of the material, the higher toughness is expected [31]. Based on Figure 8, it can be concluded that the HAZ area shows the highest toughness, which makes it the most probable area to develop necking and fail during the tensile test. This idea is also in accordance with our tensile test samples, which all necked and failed in the HAZ area.

In addition, in industrial applications, a delay between the welding process and the PWHT of the samples might happen. During this delay, welded joins age naturally. Therefore, evaluating the repeatability of the PWHTs with different durations of natural aging before artificial aging is important from an industrial perspective. The repeatability of the presented processes in Table 1 is evaluated in this part. Accordingly, the artificial aging at 160 °C for 18 h, 180 °C for 18 h, 200 °C for 2 h, 220 °C for 30 min, and 240 °C for 30 min was repeatedly done on RAFSW samples after different natural aging times, from less than one day to 21 days, prior to artificial aging. The results, shown in Table 2, demonstrate that natural aging prior to artificial aging only slightly affects the tensile properties of the joints. These results clearly validate the repeatability of the obtained artificial aging processes when natural aging happens before that.

**Table 2.** Tensile strength of the welded samples subjected to natural aging for less than one day to 21 days prior to artificial aging process conducted at different conditions based on the obtained results in the previous part of the study.

| | | UTS (MPa) after Artificial Aging at Different Conditions | | | | |
|---|---|---|---|---|---|---|
| | | At 160 (°C) for 18 h | At 180 (°C) for 18 h | At 200 (°C) for 2 h | At 220 (°C) for 30 min | At 240 (°C) for 30 min |
| **Natural aging time (days) prior to artificial aging** | 1≤ | 254 | 257 | 245 | 241 | 230 |
| | 7 | 251 | 257 | 249 | 243 | 225 |
| | 14 | 249 | 255 | 251 | 238 | 229 |
| | 21 | 250 | 252 | 246 | 238 | 230 |
| **Average UTS and standard deviation** | | **251.0 ± 1.9** | **255.2 ± 2.0** | **247.7 ± 2.4** | **240.0 ± 2.1** | **228.5 ± 2.1** |

*3.3. Solubilizing Followed by Artificial Aging (W + T6)*

Another PWHT done in this research was a solubilizing process followed by an artificial aging process called T6. In this regard, there is an interesting question among researchers on how a full W + T6 heat treatment affects a FSW joint and whether it is possible to recover the lost mechanical properties of the welds mostly via a T6 heat treatment. Though a W + T6 heat treatment is costly for most of the industrial applications, the costs could be justifiable for some applications if the W + T6 could provide better results than a single artificial aging process.

Accordingly, we have studied the impact of a standard solubilizing process at 530 °C for 1 h, followed by various artificial aging processes at 160 °C for 18 h, 180 °C for 18 h, 200 °C for 2 h, 220 °C for 30 min, and 240 °C for 30 min based on the optimized conditions of artificial aging presented in Table 1. The findings for the tensile strength of the joints are illustrated in Figure 11. It is observable that artificial aging at either 200 °C for 2 h or 180 °C for 18 h after the solubilizing process yields the best results for RAFSW joints, which are solubilized prior to artificial aging. For plain material, on the other hand, artificial aging at 180 °C for 18 h provides the best results. The results for plain material are in accordance with the reported data [26]. Figure 11 illustrates that the best condition for a RAFSW sample is not necessarily the same for the plain material. This is based on their different microstructures. A notable point for RAFSW samples is that a relatively fast aging process at 200 °C for 2 h after solubilizing at 530 °C for 1 h can achieve the best tensile strength among welds that are solubilized followed by artificial aging. This optimized time and temperature is different from the optimized aging time and temperature when there is no solubilizing process prior to artificial aging. This could be associated to the fact that a solubilizing process changes the driving force for

re-precipitation kinetics and the recrystallization of the grains. As a result, there would be differences between the precipitates' size, distribution, and morphology at the different weld zones when the welds are solubilized followed by artificial aging than just artificial aged welds [16,32–34]. The results also indicate that at relatively long aging times, the tensile properties of the base metal of the welded samples starts to degrade, as shown in Figure 11. That could contribute partly to reduction of the tensile properties of joints due to the reduction of the BM properties at aging temperatures higher than 200 °C.

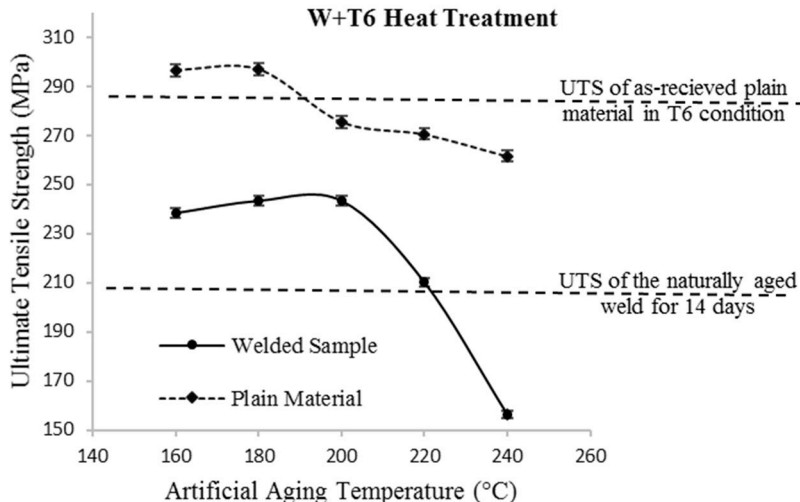

**Figure 11.** The effect of a solubilizing heat treatment followed by artificial aging at different temperatures (at optimized durations obtained from the previous section) on the tensile properties of the welded samples and plain material.

Microscopic observations demonstrate that coarsening and abnormal grain growth (AGG) have occurred within the NZ area in all samples. Figure 12 depicts the microstructure around the NZ, TMAZ, and HAZ areas of the RAFSW sample solubilized and artificially aged at 200 °C for 2 h after welding. The size of grains in the TMAZ and HAZ areas do not change considerably, while the NZ area depicts AGG. On one hand, the FSW process makes the grains in the NZ very fine compared to BM because of dynamic re-crystallization mechanisms. This causes a thermodynamic instability inside the material and encourages the material to stabilize through grain growth. On the other hand, the pinning force by precipitates against grains growth reduces during the solubilizing process after welding because of dissolution of precipitates during the solubilizing heat treatment of FSW samples [16,32–34]. As a result, AGG occurs in the NZ region during the T6 process, as can be seen in Figure 12. The level of coarsening is considerably affected by the amount of thermodynamic driving force, pinning force, the microstructure, the time and temperature of the solubilizing process, and the exact chemical composition [33]. The best W + T6 process is obtained after solubilizing at 530 °C for 1 h followed by artificial aging at 200 °C for 2 h. The joint efficiency and tensile strength for this RAFSW joint are 85% and 243 MPa, respectively.

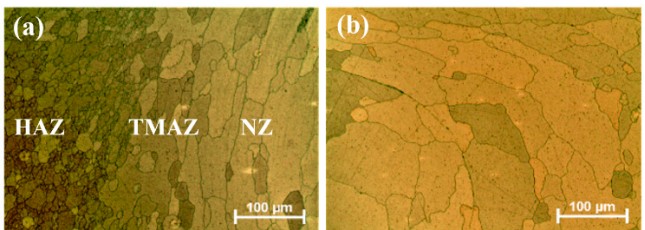

**Figure 12.** Microstructure around (**a**) and within (**b**) the nugget zone of the welded sample after W + T6 heat.

Though applying the obtained W + T6 process results in the recovery of the tensile strength of the RAFSW joint by a notable percentage, the joints have shown low ductility compared to just artificially aged samples. The low ductility in these samples can be attributed to the formation of precipitate-free zones (PFZs) and AGG within the NZ area during the T6 heat treatment [16,33,34]. All of the RAFSW samples in this research fractured within the NZ during the tensile test. That could be attributed to the fact that the fracture of these samples mostly initiates from the PFZs inside the NZ zone [16,32]. Therefore, it seems that a full W + T6 heat treatment enhances the tensile strength of the joints at the expense of their lower ductility. In summary, for RAFSW joints, conducting a solubilizing heat treatment followed by artificial aging does not yield better mechanical and physical properties than just an artificial aging process. Moreover, a solubilizing process can cause some distortions and dimensional errors, especially for complex shapes which require machining and finishing processes after heat treatment. Therefore, a single artificial aging at 220 °C for 30 min on RAFSW samples can yield not only nearly the same tensile strength but also the higher ductility than an optimized W + T6 heat treatment consisting of 1 h of solubilizing at 530 °C followed by artificial aging at 200 °C for 2 h. Moreover, a single artificial aging process on RAFSW samples is more time and cost-effective and presents lower thermal distortion rather than a full W + T6 heat treatment.

## 4. Conclusions

Based on the results obtained in this research, the main conclusions can be summarized as follows:

- A comprehensive study was done on the effect of natural aging, artificial aging, and T6 heat treatments on the mechanical and physical properties of the butt-welded bars of AA6061-T6 by a recently developed RAFSW technique using low-cost 3-axis machine tools.
- The joint efficiency of a naturally aged weld reaches 72% after 14 days.
- The optimized conditions for artificial aging were obtained as presented in Table 1. It was found that the joint efficiency of RAFSW samples reaches 90% by artificial aging at 180 °C for 18 h.
- Moreover, it was found that industrial users could take advantage of a fast artificial aging process at 220 °C for 30 min to obtain a RAFSW joint with 85% of joint efficiency and 241 MPa of tensile strength.
- In the industry, a delay is likely to happen between welding and PWHT. In this regard, the repeatability of the optimized artificial aging processes was validated for the time gaps up to 21 days between the welding and PWHT processes.
- A solubilizing heat treatment prior to artificial aging, called W + T6, at 200 °C for 2 h results in an improvement of the tensile strength of RAFSW samples up to 243 MPa. However, the welds have low ductility compared to the samples just aged artificially.
- It was shown that the artificial aging process on RAFSW samples is not only more time and cost-effective than the solubilizing followed by artificial aging process but also yields higher mechanical properties, even when it is done at a high pace.

The results provided in this research can be used as a guideline regarding the effect of various PWHT processes on AA6061-T6 butt joints made by the recently developed RAFSW technique, applicable industrially on common low-cost 3-axis CNC machines.

**Author Contributions:** M.M. and M.G. made the welds; M.M. and M.G. conceived and designed the PWHT processes and the characterization; M.M. performed the experiments, the characterization tests, and the analysis of results; M.M. wrote the manuscript; M.G. supervised the experiments and the analysis; M.G. revised the manuscript.

**Funding:** This research has been supported by PI2 Team funds.

**Acknowledgments:** The authors would like to thanks the PI2/REGAL team members who made this research possible.

**Conflicts of Interest:** The authors declare no conflict of interest.

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
