# Peer review of "Post-Weld Heat Treatment Effects on Mechanical Properties and Microstructure of AA6061-T6 Butt Joints Made by Friction Stir Welding at Right Angle (RAFSW)"

_jmmp, doi:10.3390/jmmp3020042_

Reviewer 1 Report

This is an interesting original article on investigation of mechanical properties of butt joints welded by RAFSW. The used methods are appropriate and the presentation of the results are generally clear. The manuscript is carefully written. Exceptionally, I do not have any specific comments.

Author Response

See the submitted file joined to reviewer 1.

Reviewer 2 Report

The manuscript by Momeni and Guillot descripts effect of post weld heat treatment on mechanical properties of AA6061-T6 butt joints welded by RAFSW. Artificial aging process is not only more effective on time and cost, but also yields higher mechanical properties. The reviewer has following comments:

1.    Authors emphasized that RAFSW was a low-cost FSW technique. Please show evidence.

2.    Please cite three more papers (Materials Science and Technology 35.8 (2019) 986-992, Advances in Materials Science and Engineering 2018 (2018) 4873571, and Metals 9.3 (2019) 304) at Line 49 in Page 2.

3.    Please provide FSW tool dimension.

4.    Please provide error bars in Fig. 3, Fig. 7, Fig. 8, and Fig. 11.

5.    Please mark NZ, HAZ, and TMAZ in Fig. 5(a), Fig. 9(a), and Fig. 12(a).

6.    Authors mentioned “A notable point for RAFSW samples is that a relatively fast aging process at 200 °C for 2 h can achieve the best tensile strength”. Why?

7.    Pease double check grammatical errors and polish English.

Author Response

See the file submitted to reviewer 2.

Reviewer 3 Report

Please consider changing the title into: Post Weld Heat Treatment effects on Mechanical Properties and Microstructure of AA6061-T6 Butt Joints Made by Friction Stir Welding at Right Angle. Another important suggestion would be to pay just a bit more attention to plasticity, i.e. elongation, in discussion. Also it would be beneficial if impact toughness would be at least mentioned in discussion or introduction. Other than that just few corrections are needed as those marked in the file attached. 

Author Response

See the file submitted to reviewer 3.
